# Multiple mechanisms regulate H3 acetylation of enhancers in response to thyroid hormone

Stine M. Præstholm[1], Majken S. Siersbæk[1], Ronni Nielsen[1], Xuguang Zhu[2], Anthony N. Hollenberg[3], Sheue-yann Cheng[2], Lars Grøntved[1]*

1 Department of Biochemistry and Molecular Biology, University of Southern Denmark, Odense, Denmark, 2 Laboratory of Molecular Biology, CCR, NCI, NIH, Bethesda, Maryland, United States of America, 3 Division of Endocrinology, Diabetes and Metabolism Weill Cornell Medicine, New York, New York, United States of America

* larsgr@bmb.sdu.dk

**Data Availability Statement:** All RNA-seq and H3K27Ac, H3K9Ac, H3K4me1, HDAC3, MED1, NCOR1, CBP ChIP-seq data generated in this study is deposited at Gene Expression Omnibus:

## Abstract

Hormone-dependent activation of enhancers includes histone hyperacetylation and mediator recruitment. Histone hyperacetylation is mostly explained by a bimodal switch model, where histone deacetylases (HDACs) disassociate from chromatin, and histone acetyl transferases (HATs) are recruited. This model builds on decades of research on steroid receptor regulation of transcription. Yet, the general concept of the bimodal switch model has not been rigorously tested genome wide. We have used a genomics approach to study enhancer hyperacetylation by the thyroid hormone receptor (TR), described to operate as a bimodal switch. H3 acetylation, HAT and HDAC ChIP-seq analyses of livers from hypo- and hyperthyroid wildtype, TR deficient and NCOR1 disrupted mice reveal three types of thyroid hormone (T3)-regulated enhancers. One subset of enhancers is bound by HDAC3-NCOR1 in the absence of hormone and constitutively occupy TR and HATs irrespective of T3 levels, suggesting a poised enhancer state in absence of hormone. In presence of T3, HDAC3-NCOR1 dissociates from these enhancers leading to histone hyperacetylation, suggesting a histone acetylation rheostat function of HDACs at poised enhancers. Another subset of enhancers, not occupied by HDACs, is hyperacetylated in a T3-dependent manner, where TR is recruited to chromatin together with HATs. Lastly, a subset of enhancers, is not occupied directly by TR yet requires TR for histone hyperacetylation. This indirect enhancer activation involves co-association with TR bound enhancers within super-enhancers or topological associated domains. Collectively, this demonstrates various mechanisms controlling hormone-dependent transcription and adds significant details to the otherwise simple bimodal switch model.

## Author summary

Thyroid hormone (T3) is a central regulator of growth, thermogenesis, heart rate and metabolism. In the liver T3 binds thyroid hormone receptor beta (TRβ) controlling expression of genes involved in processes such as lipid and cholesterol metabolism. The molecular mechanisms controlling TR-dependent gene regulation are centred on a

GSE128535 (https://www.ncbi.nlm.nih.gov/geo/query/acc.cgi?acc=GSE128535).

**Funding:** The research was funded by the SDU2020 initiative, the Lundbeck Foundation, the Novo Nordisk Foundation and the Danish National Research Foundation through a grant (#141) to the Center for Functional Genomics and Tissue Plasticity (ATLAS). The funders had no role in the study design, data collection and analysis, decision to publish, or preparation of the manuscript.

**Competing interests:** The authors have declared that no competing interests exist.

bimodal switch model. In the absence of T3 co-repressors bind TR reducing gene expression. When hormone binds TR, co-repressors dissociate, and co-activators are recruited inducing gene expression. This model predominates the current understanding of T3-regulated gene expression. However, only a few studies have tested this model by genome-wide approaches. We have quantified histone3 acetylation genome-wide in the liver of hypo- and hyperthyroid mice and identified gene regulatory regions regulated by T3. Probing TR and co-regulators at these regulatory regions, and analysing histone3 acetylation in mouse models for disrupted co-repressor and TR activity, reveal additional insights to the mechanisms regulating T3-dependent gene expression. We suggest a revision of the prevailing bimodal switch model which helps understanding T3-regulated gene expression in tissues such as liver. We hope that this study, together with future studies, will add new perspectives on nuclear receptor-mediated transcriptional regulation to reveal general principles.

## Introduction

Active enhancers are defined by DNase accessible chromatin, histone 3 (H3) hyperacetylation (e.g. H3K9Ac and H3K27Ac), H3K4 mono- and dimethylation (H3K4me1 and H3K4me2) and occupancy of the mediator complex [1–5]. Establishment of active enhancers has been studied extensively using nuclear hormone receptors as model systems [6]. This includes a number of key studies using TR to identify and characterize the function of transcriptional co-regulators such as, steroid receptor coactivators (SRCs) [7–9] and CBP/p300 [10,11] with HAT activity, the mediator complex [12], nuclear receptor co-repressors (NCOR1 and NCOR2) [13,14] and histone deacetylase HDAC3 [15]. In absence of ligand TR interacts with the nuclear receptor co-repressor complex (NCoR) consisting primarily of NCOR1/2, TBL1, GPS2 and HDAC3 [16–19] by a direct interaction between TR and the NCOR1 C-terminal domain [20–22]. When recruited to chromatin the unliganded TR-NCoR complex deacetylates the surrounding histone tails and facilitates active repression of transcription. In presence of thyroid hormone, the extreme C-terminal alpha helix (helix 12) of TR changes conformation, enabling the receptor to interact with transcriptional co-activators [23–26]. Recruitment of co-activators such as MED1, SRC1 and CBP/p300 leads to hyperacetylation of nearby histone tails and RNA polymerase II recruitment/activation, collectively resulting in increased expression of nearby genes [27,28]. This bimodal molecular switch has served as the canonical model describing nuclear receptor regulation of enhancer activity in the last two decades and thus is generally applied to describe TR-dependent activation of enhancers [29,30]. Yet, no studies have systematically tested this bimodal switch model in a genome-wide manner.

Importantly, in addition to direct TR binding to the genome (so called type 1 action), thyroid hormone regulates multiple processes by alternative mechanisms, including T3 actions by TR tethering to other transcription factors (TFs) (type 2 actions) and non-genomic actions in a TR-dependent (type 3 actions) and independent manner (type 4 actions) [31]. Thus, in addition to the bimodal switch model (characterized by type 1 actions), enhancer regulation by T3 may be controlled by alternative mechanisms. Moreover, previous studies have shown T3-dependent recruitment of TR to chromatin [32,33], which does not agree with the bimodal switch model. Thus, several subcategories of type 1 action of T3 may exist.

Here we used a genomics approach to test if the prevailing bimodal switch model sufficiently explains thyroid hormone regulation of enhancers. As a model system we used livers from hypo- and hyperthyroid animals and identified regulatory regions in the genome, where

histones (H3K27 and H3K9) are hyperacetylated in response to T3 (i.e. regulatory regions activated by T3). First, we defined hyperacetylated regions bound by TR and discovered considerable amount of hyperacetylated regions not occupied by TR yet required TR for activation. Next, we carefully characterized hyperacetylated regions with and without occupancy of TR using a combination of already published HiC data, extensive DNA motif analysis and ChIP-seq against co-regulators. Finally, we functionally tested the role of the NCoR complex using a mouse strain expressing a NCOR1 mutant unable to interact with TR (L-NCOR1ΔID). Collectively, this suggests that the canonical bimodal switch model does not fully explain T3-dependent activation of enhancers. We present a revised model that adds significant details to the mechanistic action of thyroid hormone signalling.

## Results

### Histone hyperacetylation of regulatory regions by thyroid hormone

Histones surrounding enhancers engaged in gene transcription are hyperacetylated at a number of lysine residues including H3K9 and H3K27 [34,35] and thus provide a molecular signature of active enhancers [2,35,36]. To characterize enhancer activation by T3 in a genome-wide manner, we initially performed a series of ChIP-seq experiments against H3K27Ac and H3K9Ac in liver tissue from hypo- (PTU treated) and hyperthyroid (PTU+T3 treated) mice. Using replicate ChIP-seq experiments (S1A–S1D Fig) we found a total of 34,533 H3K27Ac regions and 31,793 H3K9Ac regions in mouse liver tissue (S1E and S1F Fig). In combination, this revealed a total of 37,200 H3 acetylated regions (S1G Fig) with high correlation between H3K27Ac and H3K9Ac (S1H and S1I Fig). These regions varied greatly in size with a mean size of 3782bp (H3K27Ac) and 3522bp (H3K9Ac) (S1J and S1K Fig). Using these genomic coordinates, we next quantified histone acetylation in response to T3 and found that T3 treatment caused increased acetylation at both H3K9 and H3K27 at numerous regions in the genome. This includes a region near the well-described T3-regulated gene *Thrsp* (Fig 1A). To identify bona fide hyperacetylated regions in response to T3 we initially extracted regions with increased H3K27Ac and H3K9Ac in response to T3 at FDR<0.01 and log2FC>1 (S2A and S2B Fig). Next, we extracted regions hyperacetylated at both H3K27 and H3K9, collectively constituting high confidence hyperacetylated regions in the genome in response to T3 (Fig 1B; S2C Fig). Importantly, the H3K4me1 level at these regions correlated with H3K27Ac and H3K9Ac in hyperthyroid condition confirming active enhancers (S2D and S2E Fig). Moreover, the vast majority of the hyperacetylated regions are positioned in introns and intergenic regions (S2F Fig), where gene regulatory regions are most frequently positioned [37]. To test if the identified hyperacetylated regions are also present under normal physiological situations we correlated H3K27Ac under euthyroid conditions using previously published ChIP-seq data [38] and found a high correlation (S2G Fig).

The identified hyperacetylated regions covered hundreds to thousands of kilobases (S1J and S1K Fig) and, as indicated in Fig 1A, can harbour multiple DNase hypersensitive sites (DHSs) likely to bind regulatory TFs. Thus, to identify TFs involved in T3-mediated histone hyperacetylation we extracted all DHSs within these regions (total of 3778 DHSs) and performed *de novo* DNA motif analysis. This showed that the TR response element (direct repeat separated by four nucleotides, DR4) was the most significantly enriched motif, indicating direct TR binding to response elements within hyperacetylated regions (Fig 1C). Yet, the TR response element was only found in 17% of the 3778 DHSs, representing 37% of the hyperacetylated regions. In agreement, integration of TR ChIP-seq data [32] revealed that less than 40% of the hyperacetylated regions contain at least one TR binding site (TRBS) (Fig 1D, n = 601). Quantification of H3K27Ac and H3K9Ac at the TR bound, and unbound regions showed that the histone

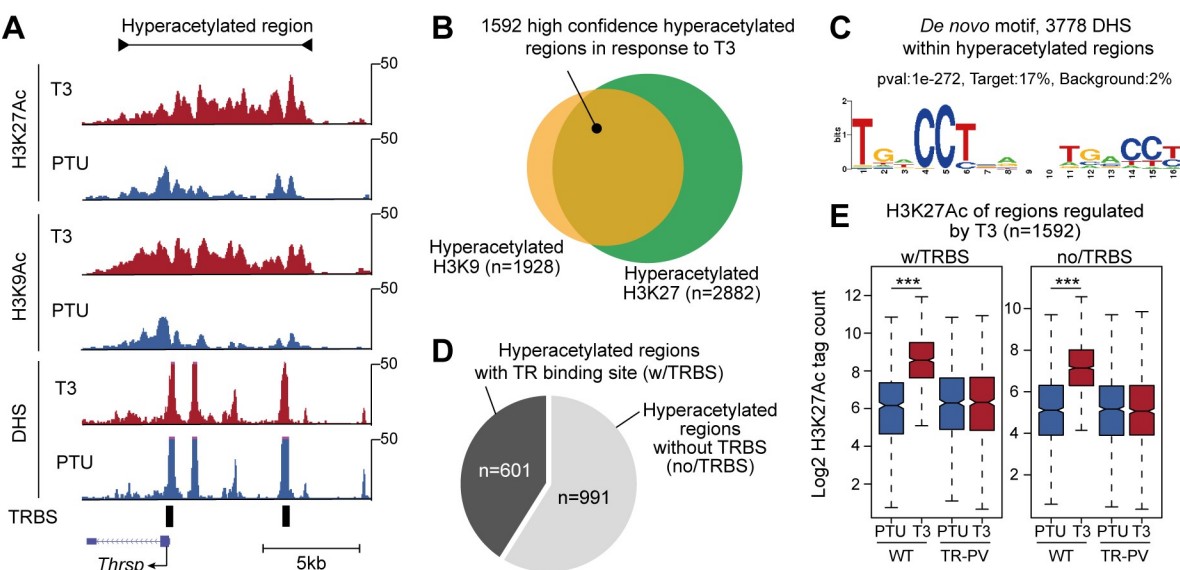

**Fig 1. Identification of T3-regulated hyperacetylated regions in the liver.** (A) Example of increased H3K9 and H3K27 acetylation at DHSs near the *Thrsp* gene. TR binding sites (TRBSs) identified from TR ChIP-seq data are indicated as black bars. The identified hyperacetylated region is displayed above density graphs of sequenced tags. (B) 1592 regions hyperacetylated at both H3K9 and H3K27 constitute high confidence hyperacetylated regions in hyperthyroid condition. (C) *De novo* motif analysis of 3778 DHSs within hyperacetylated regions. The most enriched motif is shown. (D) Relative amount of hyperacetylated regions with at least one TRBS based on TR ChIP-seq data. (E) T3-induced H3K27Ac in mice expressing TR-PV mutant lacking a functional activating helix 12. Statistical difference was determined by a Wilcoxon Signed Rank Test, ***p<0.001.

acetylation level in general was lower at hyperacetylated regions not bound by TR (S2H Fig). This suggests that occupancy of TR is associated with a general more robust H3 acetylation.

The absence of TR binding in the majority of hyperacetylated regions, suggests a considerable number of regions hyperacetylated by T3 in a TR-independent manner. To test this, we replicated the H3K27Ac ChIP-seq experiments in a TRβPV/PV (TR-PV) mutant mouse model [39], where TR is functional inactive as a result of frame-shift mutation in the ligand-binding domain [40]. Interestingly, using this model we observed a complete lack of T3-induced hyperacetylation (Fig 1E), indicating that TR in part regulates histone hyperacetylation without being directly associated with the hyperacetylated region. This could be mediated by T3-regulated expression of TFs that in turn facilitate histone hyperacetylation (i.e. secondary TFs) or T3 control of signalling pathways potentially regulating activity of specific downstream TFs (type 3 actions). For example, T3 has been shown to regulate the activity of PI3K [41]. Alternatively, enhancers not bound by TR may be hyperacetylated through close proximity of TRBSs in the three-dimensional space.

To address secondary genomic effects of T3 we quantified genome wide H3K27Ac in liver from hypothyroid mice treated acutely with T3 for 2 or 6 hours (h). The vast majority (~85%) of hyperacetylated regions with a TRBS had increased H3K27Ac after 2h of T3 treatment (S3A–S3C Fig), indicating predominately direct genomic effects of TR. Interesting, most (~60%) hyperacetylated regions without a TRBS also showed increased H3K27Ac in response to acute T3 treatment (S3D and S3E Fig), demonstrating that a significant number of regions not occupied by TRBS were hyperacetylated by the same kinetic profile as regions with a TRBS. This suggests a potential co-regulatory mechanism. Notably, there was clearly less regions without a TRBS with increased H3K27Ac in response to acute T3 treatment compared to regions with a TRBS (S3C Fig), suggesting that some hyperacetylated regions may be regulated by secondary TFs.

## Putative T3-regulated TFs controlling histone hyperacetylation

To identify potential secondary TFs involved in T3-regulated histone hyperacetylation we applied a machine-learning approach IMAGE [42] using H3K27Ac and H3K9Ac quantified at DHSs combined with hepatic RNA-seq data from hypo- and hyperthyroid animals. Out of 1611 curated TF binding sites (DNA motifs), IMAGE identified 51 and 50 DNA motifs contributing to H3K27 and H3K9 hyperacetylation (p<0.01), respectively (S4A and S4B Fig). A total of 31 DNA motifs contributed to both H3K27 and H3K9 acetylation (S4C and S4D Fig) and correlation of the position weight matrices showed that DNA motifs resembling the DR4 motif displayed the most significant differential motif activity in response to T3 (S4E Fig). Moreover, a number of additional DNA motifs showed significant contribution to T3-induced H3K27Ac and H3K9Ac including motifs shown to bind transcription factors such PPARD, ESR1, CEBPB and SREBF2/SREBP2 (S4D and S4E Fig). However, quantification of the motif scores within DHSs in the TR bound and unbound hyperacetylated regions, suggested that none of the individual non-DR4 motifs contributed significantly to histone hyperacetylation of regions without a TRBS (S4E Fig, right). In contrast, we did observe differential DR4 motif score enrichment in agreement with differential TR occupancy. Moreover, we performed *de novo* motif analysis of DHSs associated with hyperacetylated regions with and without TRBSs. In agreement with presence of TRBSs we identified the DR4 motif exclusively in the DHSs associated with regions occupied by TR (S4F Fig). In addition, we identified a number of additional motifs, however none of these showed differential motif score enrichment comparing regions with and without TRBS (S4F Fig). Thus, T3-dependent hyperacetylation of regions not bound by TR is likely not explained by occupancy of individual secondary transcription factors.

## Higher-order organization of hyperacetylated enhancers

Organization of individual enhancers into co-regulatory hubs of enhancer-clusters is central for balanced mRNA transcription in response to various signals [43]. This higher-order organization has been described at different levels such as relatively close proximity (10-15kb) of individual enhancers into super-enhancers (SE) and long-range interaction of enhancers spanning hundreds of kilobases within topological associated domains (TADs) [44,45]. Thus, T3-dependent hyperacetylation of enhancers may arise from enhancer co-regulation within interacting clusters of individual enhancers.

Using the H3K27Ac ChIP-seq data we identified 688 SEs in hyperthyroid condition, with a mean size of 24kb (Fig 2A) and about seven times the average size of hyperacetylated regions. More than 22% of the SEs contained T3 hyperacetylated regions of which two third contained hyperacetylated regions with TRBSs (Fig 2B). Close to 60% of the SEs containing hyperacetylated regions without a TRBS were co-occupied by hyperacetylated regions with a TRBS, which was significantly higher compared to the frequency of co-occupancy by random H3 acetylated regions (Fig 2C). Moreover, close to 7% of 991 individual hyperacetylated enhancers, not occupied by TR, were positioned in a SE also containing hyperacetylated regions with a TRBS (Fig 2D). Although this only constitutes a small fraction of the non-TR occupied hyperacetylated enhancers it represents more than a threefold enrichment compared to randomly selected histone acetylated regions. Remarkably, hyperacetylated regions without a TRBS co-occurring in SE with a TRBS were more pronouncedly H3K27acetylated in response to acute (2h) T3 treatment compared to regions not in a SE with TRBSs (Fig 2E). This suggests that hyperacetylation of some non-TR bound enhancers may arise from enhancer co-regulation within SEs. Hyperacetylated regions near the *Proca1* gene represent one example (Fig 2F).

SEs are composed of enhancer constituents spaced 10-15kb and thus fail to describe long-range enhancer interactions spanning hundreds of kb. To analyse the potential impact of long-

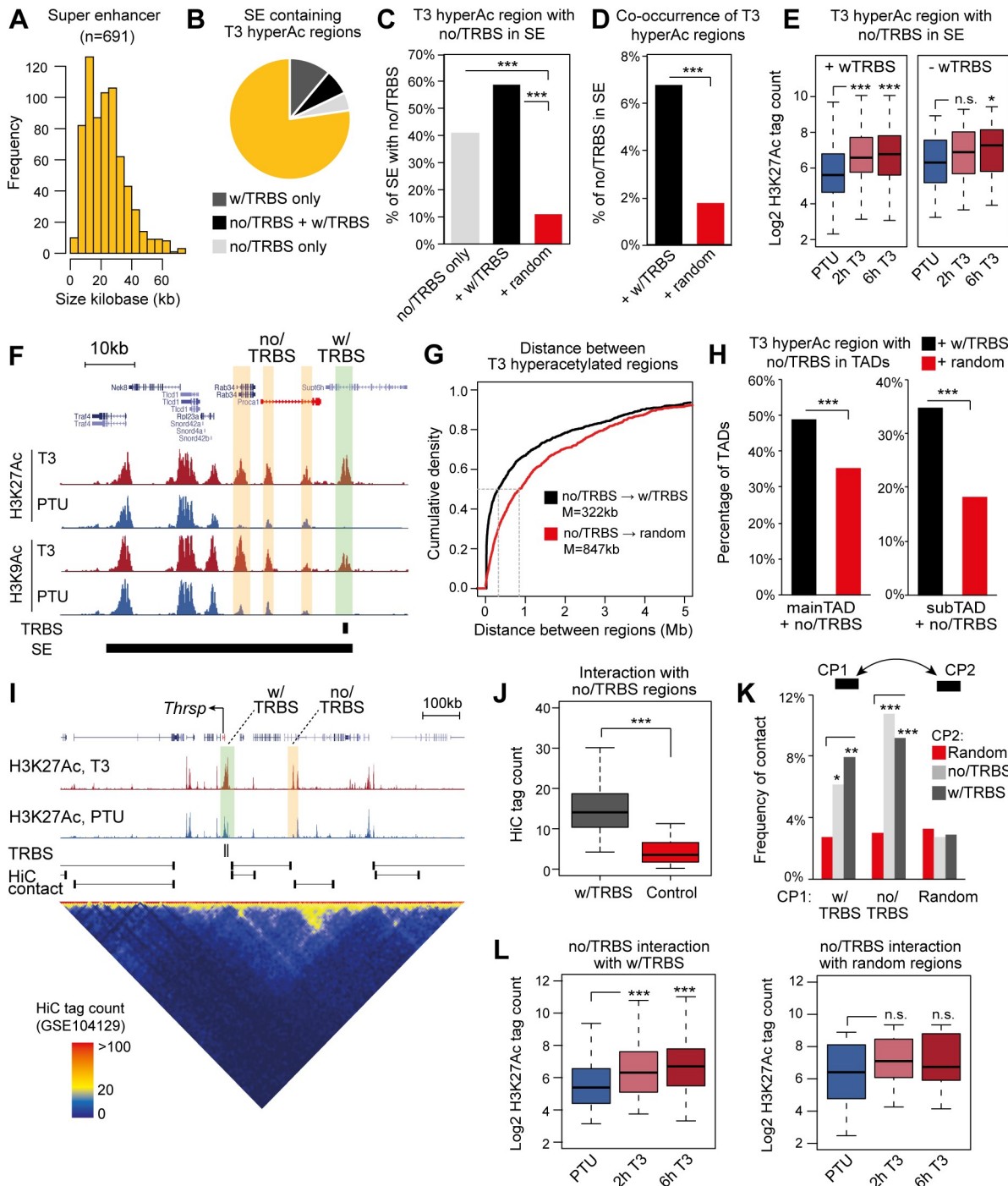

**Fig 2. Configuration of hyperacetylated enhancers in SEs and TADs.** Hyperacetylated regions were divided into regions with TRBS (w/ TRBS) and without TRBS (no/TRBS). (A) Size distribution of SE in hyperthyroid condition based on H3K27Ac. (B) Fraction of SE containing T3-responsive hyperacetylated enhancers with TRBS and without TRBS. (C) Fraction of SEs containing a hyperacetylated region without a TRBS together with regions containing a TRBS. ***Fisher's exact test p< 0.001 using 600 random selected histone acetylated regions. (D) Fraction of hyperacetylated regions without a TRBS that co-occurs in SE with hyperacetylated regions with a TRBS. ***Fisher's exact test p< 0.001 using 600 random selected histone acetylated regions. (E) Acute H3K27Ac of regions without a TRBS in a SE with and without hyperacetylated regions containing a TRBS. Statistical difference was determined by a Wilcoxon Signed Rank Test, *p<0.05, ***p<0.001. Non significant is marked by n.s. (F) Example of a SE near the T3-regulated *Proca1* gene. (G) Distance between hyperacetylated regions w/TRBS and no/TRBS. The median [M] distance is indicated for no/TRBS to w/TRBS containing regions and no/TRBS to random regions. (H) Fraction of hyperacetylated regions without TRBSs within TADs containing a TRBS. ***Fisher's exact test p< 0.001 using 600 random selected histone acetylated regions. (I) Example of HiC contacts in a TAD harbouring the *Thrsp* gene. (J) HiC tag count of all

detected interactions between hyperacetylated regions with and without TRBS. Interaction between regions in the opposite direction was quantified as a background control. Statistical difference was determined by a Wilcoxon Signed Rank Test (n = 56), ***p<0.001. (K) Frequency of hyperacetylated regions contacting other hyperacetylated regions. Fisher's exact test ***p<0.001, **p< 0.01, *p<0.05 using 365 random selected histone acetylated regions. (L) Acute H3K27Ac of regions without a TRBS interacting with regions with a TRBS or random selected regions. Statistical difference was determined by a Wilcoxon Signed Rank Test, *p<0.05, ***p<0.001. Non-significant is marked by n.s.

range interactions between enhancers we first mapped the distance between hyperacetylated regions not bound by TR and regions bound by TR and compared this with the distance to random histone acetylated regions. This showed that most hyperacetylated regions not bound by TR were on average positioned hundreds of kb from hyperacetylated regions bound by TR (Fig 2G). Interestingly, the hyperacetylated regions without TRBSs were positioned closer to hyperacetylated with TRBS compared to random (Fig 2G), suggesting that the relative distance between hyperacetylated regions could have a regulatory role.

Interactions between regulatory regions are often confined in TADs [43], suggesting that TAD organization in the liver may confine T3 co-regulated regions. To test this, we quantified histone acetylated regions within previously mapped TADs and subTADs in mouse liver tissue (TADs defined by HiC data and subTADs by sub-megabase structures of different lengths within the TADs flanked by CTCF, cohesin and occupied by MED1) [46]. This showed that hyperacetylated regions without TRBSs are more frequently positioned in the same TAD as hyperacetylated regions with TRBS compared to randomly selected regions (Fig 2H).

To map actual interactions between hyperacetylated regions we used previously published deep sequenced HiC libraries from liver tissue [46] and scored specific interactions between hyperacetylated regions with and without TRBS. Combining all HiC libraries from two different circadian time points (ZT10 and ZT22), we scored more than 14,000 total interactions with a median distance of 190kb (S5A Fig). Extracting HiC contact regions overlapping with hyperacetylated regions indicated smaller interacting distances (~145kb), yet similar to randomly selected H3 acetylated regions in the genome (S5B Fig). For example, we observed an interaction between a hyperacetylated region more than 100kb upstream of the TR bound hyperacetylated region near *Thrsp* (Fig 2I). Similar long-distance interactions between hyperacetylated regions were observed near other T3-regulated genes such as *Ginm1* and *Glul* (S5C and S5D Fig). Next, we selected all detected interactions between hyperacetylated regions with and without TRBSs (n = 56, 6% of all hyperacetylated regions without TRBS) and compared HiC signals for these interactions with HiC of interactions in the opposite direction (Fig 2J). This confirms that detected contacts were significantly higher than background. Moreover, we found that interactions between hyperacetylated regions with TRBS and without TRBS are as frequent as interactions between regions with TRBS (Fig 2K). Importantly, this level of interaction was lower between randomly selected regions (Fig 2K), suggesting that a subset of the hyperacetylated regions not bound by TR were hyperacetylated as a result of interaction with regions occupied by TR. In agreement, we observed that hyperacetylated regions not bound by TR had a higher tendency to be acutely H3K27acetylated in response to T3 (2h) if they interacted with regions bound by TR compared to random selected regions (Fig 2L).

Collectively, these data suggest that a subset of T3 hyperacetylated regions without a TRBS can be acetylated by presence in SEs with TRBSs and/or by direct interaction with regions containing a TRBS. Thus, the higher order chromatin structure allows enhancer activation by T3 in a TR dependent manner without direct TR interaction with the enhancer.

## Distinct co-regulator recruitment to TRBSs suggests several mechanisms for enhancer regulation

The canonical model describing T3-regulated histone acetylation of enhancers infers that unliganded TR recruits HDAC3 to chromatin and maintains local histone hypoacetylation. Binding of T3 results in reduced HDAC3 occupancy and increased HAT recruitment, collectively leading to histone hyperacetylation. To test this model on a genome-wide scale, we first profiled HDAC3 by ChIP-seq and mapped HDAC3 binding sites to hyperacetylated regions. In agreement with direct action of TR, HDAC3 occupancy was preferentially enriched at T3-regulated enhancers bound by TR compared to T3-regulated enhancers not occupied by TR (Fig 3A). Interestingly, in hypothyroid condition HDAC3 occupied a minority of the hyperacetylated regions with a TRBS (Fig 3B), suggesting that differential occupancy of HDAC3 only partly explains T3-induced histone acetylation.

To further characterize the hyperacetylated enhancers associated with TRBSs (type 1 actions of T3) we initially extracted all TRBSs within hyperacetylated regions (n = 720). We subsequently quantified occupancy of different co-regulators known to interact with unliganded TR (HDAC3 and NCOR1) and ligand bound TR (CBP, p300, SRC1 and MED1). Note that p300 and SRC1 were only quantified at selected genomic regions due to low ChIP-seq signal to noise ratio for these factors. Based on HDAC3 ChIP-seq tag density in hypothyroid condition we binned the TRBSs in high and low HDAC3 bound regions, referred to as type 1A and type 1B TRBS, respectively (Fig 3C and 3D). This clearly showed that only a subset of TRBSs were occupied by the histone deacetylase in absence of hormone, also illustrated by the frequency of identified HDAC3 peaks in hypothyroid condition (S6A Fig). Moreover, the pattern of HDAC3 occupancy was confirmed by NCOR1 ChIP-seq (Fig 3C and 3D). Importantly, acute T3-induced H3K27Ac of type 1A and type 1B showed a similar pattern, where the majority of TRBSs were hyperacetylated within two hours of T3 treatment (S6B Fig). Collectively, this indicates that the NCoR complex is not actively engaged in histone hypoacetylation at all putative TRBSs, suggesting that the canonical bimodal switch model does not fully explain enhancer activation by TR bound directly to DNA.

In hypothyroid condition TR primarily occupied type 1A TRBSs (S6A Fig), suggesting that TR potentially recruits the NCoR complex to chromatin to maintain histone hypoacetylation specifically at type 1A sites. Interestingly, we found that the histone acetyltransferases CBP, p300 and SRC1 occupied these regions in both hypo- and hyperthyroid condition (Fig 3C–3E), suggesting that occupancy of HATs is not obligatory associated with hyperacetylated chromatin but can also be found at hypoacetylated regions. Similar findings have been reported for poised enhancers during embryonic development, where CBP can interact with hypoacetylated chromatin in ES cells poised for activation during differentiation [5,47]. Thus, type 1A TRBSs may represent poised hepatic enhancers in hypothyroid condition. Accordingly, we found relatively high H3K4me1 levels at these enhancers (Fig 3D), also indicative of poised enhancer status [2]. Upon activation by T3, HDAC3 and NCOR1 occupancy was reduced and CBP, p300 and SRC1 occupancy remained unchanged (Fig 3C–3E), suggesting that T3-induced H3 hyperacetylation at these poised TRBSs is mediated by decreased occupancy of HDACs rather than increased occupancy of HATs.

Focusing on the type 1B TRBSs showed that occupancy of CBP, p300 and SRC-1, as well as HDAC3 and NCOR1, was increased in response to T3 (Fig 3C–3E). Together with the more pronounced gain in chromatin accessibility and increased H3K4me1 (Fig 3D), suggest that this group of TRBSs is inaccessible and inactive in hypothyroid conditions and T3 treatment establishes accessible and active enhancers. TR occupancy at type 1B TRBSs was relatively low in hypothyroid condition (S6A Fig), indicating that TR is not involved in maintenance of a

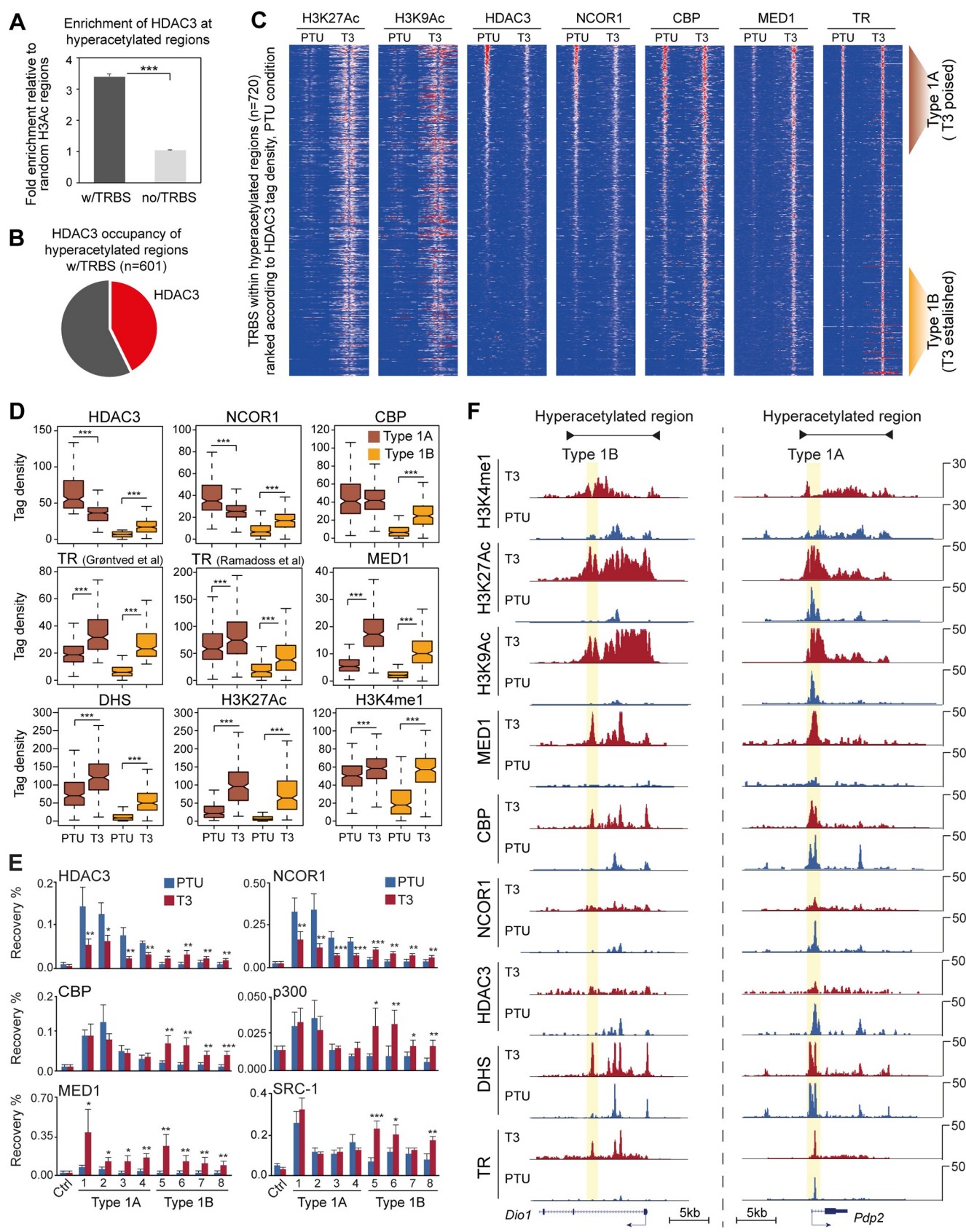

**Fig 3. Occupancy of transcriptional co-regulators at TRBSs within T3 hyperacetylated regions.** (A) Enrichment of HDAC3 occupancy within hyperacetylated regions compared to random selected H3 acetylated regions. Statistical test was performed by a Wilcoxon Signed Rank Test, ***p<0.001. (B) Fraction of hyperacetylated regions containing a TRBS (w/TRBS) occupied by HDAC3. (C) Heatmap illustrating H3K27Ac, H3K9Ac, co-regulator and TR occupancy at TRBSs within hyperacetylated regions (n = 720). TRBSs are sorted according to HDAC3 occupancy in hypothyroid condition. Type 1A TRBSs, poised enhancers, are defined by high occupancy of HDAC3, CBP and H3K4me1. Type 1B TRBSs, T3-established enhancers, are defined by low occupancy of HDAC3, CBP and H3K4me1 in absence of T3. (D) ChIP-seq tag density of histone modifications, DHS, TR and co-regulators at TRBSs associated with poised (type 1A) and T3-established enhancers (type 1B). Statistical difference was determined by a Wilcoxon Signed Rank Test. (E) ChIP-qPCR for four type 1A TRBSs (1–4) and four type 1B TRBSs (5–8). Genomic coordinates can be found in S2 Table. Statistical difference was determined by Student's t-test (n = 4), *p<0.05, **p<0.01 and ***p<0.001. (F) Example of co-regulator occupancy of hyperacetylated regions near T3-regulated genes *Dio1* and *Pdp2*.

hypoacetylated state in absence of T3. However, as TR together with HATs were strongly recruited to these sites in presence of T3 (Fig 3D and 3E), suggests that hyperacetylation is mediated by TR and HAT recruitment. Importantly, using an unrelated TR ChIP-seq data set [33] confirmed T3-dependent recruitment of TR to chromatin at type 1B TRBSs (Fig 3D). Although this unrelated TR ChIP-seq dataset was based on tagged TR overexpression in liver, potentially overestimating TR occupancy, it agrees well with the ChIP-seq data probing endogenous TR. Interestingly, ChIP-seq against MED1 suggests that the mediator complex was generally recruited to hyperacetylated TRBS in a ligand-dependent manner (Fig 3C–3E). Thus, whereas the HATs can occupy hypoacetylated chromatin, MED1 recruitment is strictly associated with ligand bound TR at hyperacetylated chromatin. This agrees well with previous studies showing that MED1 is required for full TR activity [48] and histone hyperacetylation precedes MED1 [49].

Taken together these data indicate that type 1 actions of TR should be subdivided into at least two categories. The high NCOR1-HDAC3 occupied TRBSs (type 1A) are poised enhancers, where the NCoR complex may function as a histone acetylation rheostat regulating the overall level of histone acetylation deposited by HATs. In contrast, activation of low NCOR1-HDAC3 occupied TRBSs (type 1B) is established in a T3-dependent manner by recruitment of TR and HATs leading to increased histone acetylation. Regions near the well-known TR target genes, *Pdp2* and *Dio1*, represent examples of poised (type 1A) and established (type 1B) TRBSs, respectively (Fig 3F).

## Disrupted NCoR-TR interaction leads to increased histone acetylation at a subset of TR-regulated enhancers

The poised enhancer model described above infers that disruption of NCoR recruitment to chromatin will result in increased histone acetylation at unliganded type 1A TRBSs and not type 1B TRBSs. To test this, we used the L-NCOR1ΔID mouse model. These mice express a liver-specific NCOR1 mutant lacking two TR interaction domains resulting in disruption of NCoR interaction with TR [50]. In agreement, we observed considerable decreased HDAC3 occupancy of TRBSs in the livers from L-NCOR1ΔID mice compared to WT mice in hypothyroid condition (Fig 4A and S6C and S6D Fig). Interestingly, reduced HDAC3 occupancy was also observed for a number of non-TRBSs, suggesting that the deleted domains may be important for NCOR1-HDAC3 recruitment to other transcriptional regulators than TR. With focused attention on all T3-regulated regions harbouring a TRBS (defined in Fig 1D) we found that 33% of the 601 T3-regulated enhancers became hyperacetylated at H3K9 and H3K27 in NCOR1ΔID compared to WT (Fig 4B and 4C). Importantly, this indicates that histone hypoacetylation at the majority of TRBSs is not dependent on NCoR occupancy, in agreement with previous findings from a NCOR1 KO model [38]. This supports the observation that only a fraction of TRBSs is occupied by NCOR1-HDAC3 in hypothyroid condition.

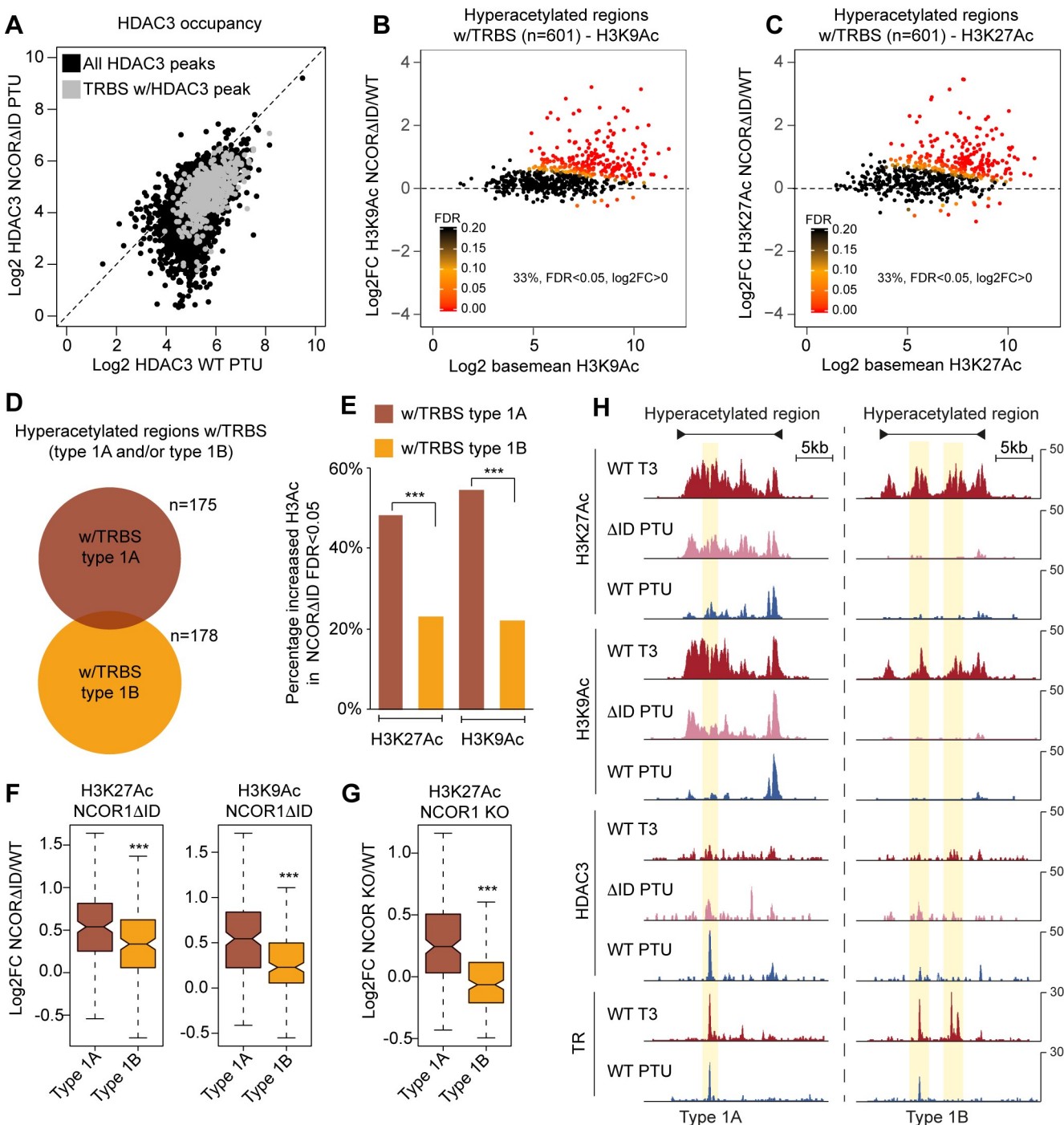

**Fig 4. Histone acetylation of TR occupied regions due to disrupted interaction between TR and NCoR.** (A) HDAC3 occupancy of chromatin in WT mice and mice expressing a mutated NCOR1 (NCOR1ΔID) unable to interact with TR. TRBSs occupied by HDAC3 are marked with grey. (B) Differential H3K9Ac of hyperacetylated regions in WT hypothyroid mice compared to hypothyroid NCOR1ΔID mice. (C) Differential H3K27Ac of hyperacetylated regions in WT hypothyroid mice compared to hypothyroid NCOR1ΔID mice. (D) Genomic overlap of hyperacetylated regions occupied by TR (w/TRBS) and low or high occupancy of NCOR1-HDAC3. (E) Fraction of hyperacetylated regions with increased H3K9Ac and H3K27Ac in livers from hypothyroid L-NCOR1ΔID mice compared to hypothyroid WT mice. ***Fisher's exact test p< 0.001. (F) Fold change of H3K9/27Ac at hyperacetylated regions occupied by TR and associated with high and low occupancy of NCOR1-HDAC3 in livers from hypothyroid L-NCOR1ΔID mice compared to hypothyroid WT mice. Statistical difference was determined by a Wilcoxon Signed Rank Test, ***p<0.001. (G) Fold change of H3K27Ac at hyperacetylated regions occupied by TR and associated with high and low occupancy of NCOR1-HDAC3 in livers from hypothyroid NCOR1 KO mice compared to hypothyroid WT mice. Statistical difference was determined by a Wilcoxon Signed Rank Test, ***p<0.001. (H) Examples of hyperacetylated regions occupied by type 1A TRBS (left, chr16:21,817,000–21,842,000) and type 1B TRBS (right, chr2:167,527,000–167,558,000).

To further characterize the function of NCoR we isolated hyperacetylated regions occupied by type 1A and type 1B TRBSs (Fig 4D). In agreement with the proposed poised enhancer model, we found that the percentage of the T3-regulated regions containing type 1A TRBSs showed more pronounced increased acetylation in the NCOR1 mutant compared to regions with type 1B TRBSs (Fig 4E). Moreover, the overall fold change of histone acetylation was more pronounced at regions with type 1A TRBSs compared to regions with type 1B TRBSs (Fig 4F). To validate this, we used previously published H3K27Ac ChIP-seq data from hypo-thyroid liver-specific NCOR1 KO mice [38] and found increased histone acetylation in NCOR1 KO specifically at regions with type 1A TRBSs (Fig 4G). Fig 4H illustrates examples of hyperacetylated regions with type 1A and type 1B TRBSs.

## Derepressed TRBS correlates with derepressed gene expression

To link histone hyperacetylation to gene expression we initially performed RNA-seq in hypo- and hyperthyroid WT mice and identified 1015 genes induced by T3 treatment (Fig 5A). Of these genes, 472 (47%) were associated with at least one hyperacetylated region within 100kb of the TSS (Fig 5A). In contrast, 2% of T3-repressed genes were associated with hyperacety-lated regions, indicating a strong correlation between T3-induced histone hyperacetylation and T3-induced gene expression (Fisher's exact test $< 2.2e-16$). Out of the 472 genes associated with hyperacetylated regions, about a third was associated with nearby hyperacetylated regions without a TRBS. And two thirds of the genes were associated with hyperacetylated regions with a TRBS (Fig 5B), suggesting a gene regulatory function for both types of hyperace-tylated enhancers defined in Fig 1. Interestingly, gene expression in hypothyroid condition as well as in presence of T3 was lower for T3-regulated genes associated with hyperacetylated regions without a TRBS compared to genes associated with hyperacetylated regions with a TRBS (Fig 5C). This agrees well with the general lower acetylation level of T3-regulated regions not bound by TR (S2H Fig), consistent with previous findings reporting direct correlation between gene expression and the level of nearby histone acetylation [2,51].

To specifically test if NCoR recruited to TRBS in hypothyroid condition is involved in nearby gene expression, we performed RNA-seq from liver of hypothyroid L-NCOR1ΔID mice and analysed differential gene expression compared to hypothyroid WT mice. Strikingly, only a minority (10%) of the T3-induced genes were affected in mice expressing NCOR1ΔID (Fig 5D). This suggests that chromatin bound NCOR1-HDAC3 only controls a small subset of T3-regulated genes in hypothyroid condition which agrees well with previous studies of NCOR1 KO mice [38]. To specifically analyse the effect of the NCOR1ΔID mutant on TRBSs associated with the NCoR complex (as defined in Fig 3C) we extracted genes associated with type 1A TRBS (n = 100 genes) and genes associated with type 1B TRBS (n = 85 genes) (Fig 5E). Quantification of genes regulated by NCOR1ΔID within these two groups of genes showed that 16 of the type 1A TRBS associated genes (16%) were induced in presence of NCOR1ΔID (Fig 5F), whereas this could only be observed for 5 genes (6%) associated with regions with type 1B TRBSs (Fig 5G, p-value = 0.037, Fisher's exact test). This indicates that the NCoR complex is more important for the repression of genes associated with a TRBS with high NCoR occupancy than at TRBS with low NCoR occupancy. This is further illustrated by comparing the fold enrichment of genes regulated in hypothyroid NCOR1ΔID mice relative to sets of 1000 random selected genes (Fig 5H). Genes associated with nearby occupancy of NCOR1-H-DAC3 (type 1A TRBS) have a higher tendency to be deregulated in the NCOR1ΔID mutant compared to genes not associated with NCOR1-HDAC3 (type 1B TRBS) or randomly selected genes.

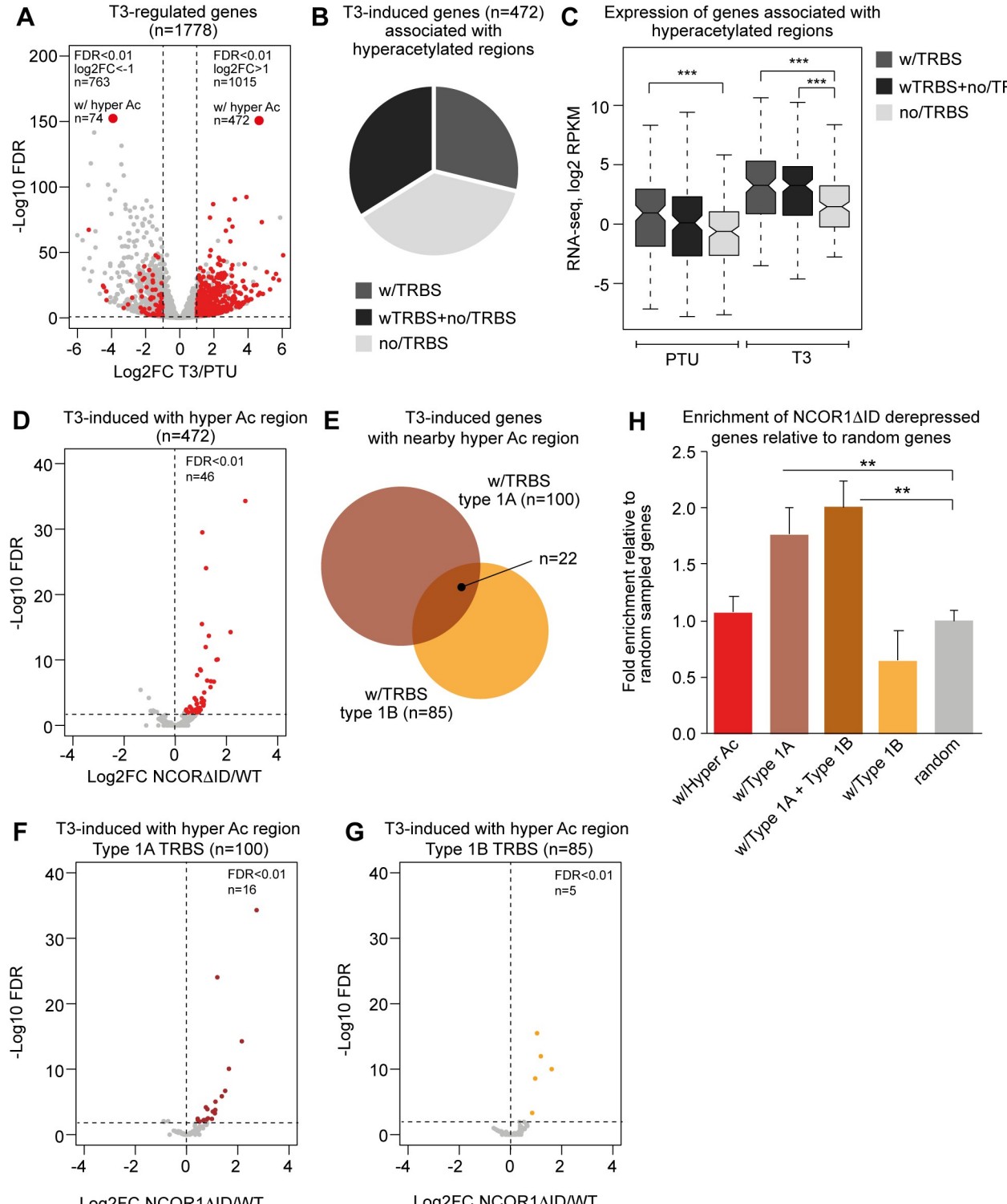

**Fig 5. Expression of genes associated with hyperacetylated regions with and without occupancy of HDAC3.** (A) Differentially expressed genes in hyperthyroid condition. Genes associated with hyperacetylated regions are marked by red. Genes are scored positive if one or more hyperacetylated regions are within 100kb of the transcriptional start site of the gene. The number of induced and repressed genes at FDR<0.01 and log2FC of one are indicated. (B) Fraction of T3-induced genes with nearby hyperacetylated regions with and without TRBSs. (C) Expression of genes associated with hyperacetylated regions as sequenced reads per kilo base per million total reads (RPKM). Statistical difference was determined by a Wilcoxon Signed Rank Test, ***p<0.001. (D) Differential expression of T3-induced genes associated with hyperacetylated regions in hypothyroid WT mice compared to

hypothyroid L-NCOR1ΔID mice. Red marks induced genes in L-NCOR1ΔID mice at FDR<0.01 and log2FC>0. (E) Number of T3-induced genes associated with hyperacetylated type 1A (brown) or type 1B (orange) TRBSs. (F) Differential expression of T3-induced genes associated with type 1A TRBSs in hypothyroid WT mice compared to hypothyroid L-NCOR1ΔID mice. Brown marks induced genes in L-NCOR1ΔID mice at FDR<0.01 and log2FC>0. (G) Differential expression of T3-induced genes associated with hyperacetylated regions occupied by type 1B TRBSs in hypothyroid WT compared to hypothyroid L-NCOR1ΔID mice. Yellow marks induced genes in L-NCOR1ΔID mice at FDR<0.01 and log2FC> 0. (H) Enrichment of genes induced in hypothyroid L-NCOR1ΔID mice relative to random selected genes. Genes are binned according to association with hyperacetylated type 1A and/or type 1B TRBSs. Ten different sets of 1000 randomly selected genes from all annotated genes were used as control. Statistical difference is calculated using a Student's t-test (n = 10), **p<0.01.

## Discussion

Here we used a genomics approach to study enhancer activation by T3 in liver tissue. We focused exclusively on enhancers hyperacetylated in response to hormone treatment and found that enhancers are activated by several possible mechanisms depending on the chromatin context. These include a) hyperacetylation of poised enhancers as a result of HDAC3 loss and not increased HAT recruitment to the enhancer, b) hyperacetylation of dormant enhancers as a result of TR and HAT recruitment to the enhancer and c) hyperacetylation of enhancers not bound by TR in part as a result of higher-order chromatin structures. Importantly, hyperacetylation at all T3-regulated enhancers require functional TR expression. Moreover, we observed that MED1 is generally ligand dependently recruited to TR-bound hyperacetylated enhancers, suggesting that mediator recruitment is a hallmark of hyperacetylated TR occupied enhancers in response to T3.

Hypoacetylated enhancers, associated with H3K4me1, hypersensitive to DNase and co-occupied by HATs and HDACs, are inactive yet poised for activation when the right molecular signal arrives [1]. We show that poised enhancers bound by unliganded TR are co-occupied by HATs and the NCoR complex capable of acetylating and deacetylating lysines at histone tails, respectively. HATs do not interact directly with unliganded TR [10,11] but interaction with TR occupied enhancers could be mediated by RXR possibly though SRCs [52–54] or other transcription factors occupying the enhancers (Fig 6A). Interestingly, although HATs interact with poised enhancers overall histone hypoacetylation is maintained either by selective control of the HAT activity or by actively deacetylating histones by NCoR recruitment. We show that disruption of NCoR interaction with TR leads to hyperacetylation of a subset of TR bound poised enhancers. However, it is important to note that only half of the poised enhancers, occupied by high levels of NCOR1-HDAC3, were hyperacetylated as a result of NCOR1-HDAC3 depletion (Fig 4E). This argues for additional NCOR1 independent mechanisms for histone hypoacetylation as suggested previously [38]. Thus, additional histone deacetylase complexes, such as SIN3A complexes containing HDAC1 or HDAC2, may be involved in active histone hypoacetylation at TRBSs when NCOR1-HDAC3 is depleted [55,56]. An alternative explanation may include differential activity of HATs at the NCOR1-HDAC3 depleted TRBS in hypothyroid condition. A prerequisite for hyperacetylation, as a result of NCoR depletion, is increased activity of pre-occupied or recruited HATs. Enhancer selective CBP activity has been demonstrated in several studies and involves multiple mechanisms including eRNAs [57] and posttranslational modifications [58–60]. Thus, increased histone acetylation upon NCoR depletions may be determined by a combination of HDAC depletion and residual HAT activity at the TRBSs in a chromatin context dependent manner.

Interestingly, we found a large fraction of TRBSs not associated with NCOR1-HDAC3 in hypothyroid condition. Compared to the poised TRBSs, these regions were less likely to be hyperacetylated in response to NCOR1-HDAC3 depletion. Moreover, these regions recruited TR and HATs in presence of T3, correlating with histone hyperacetylation. In addition, both DNase accessibility and H3K4me1 were dramatically increased at these regions in response to

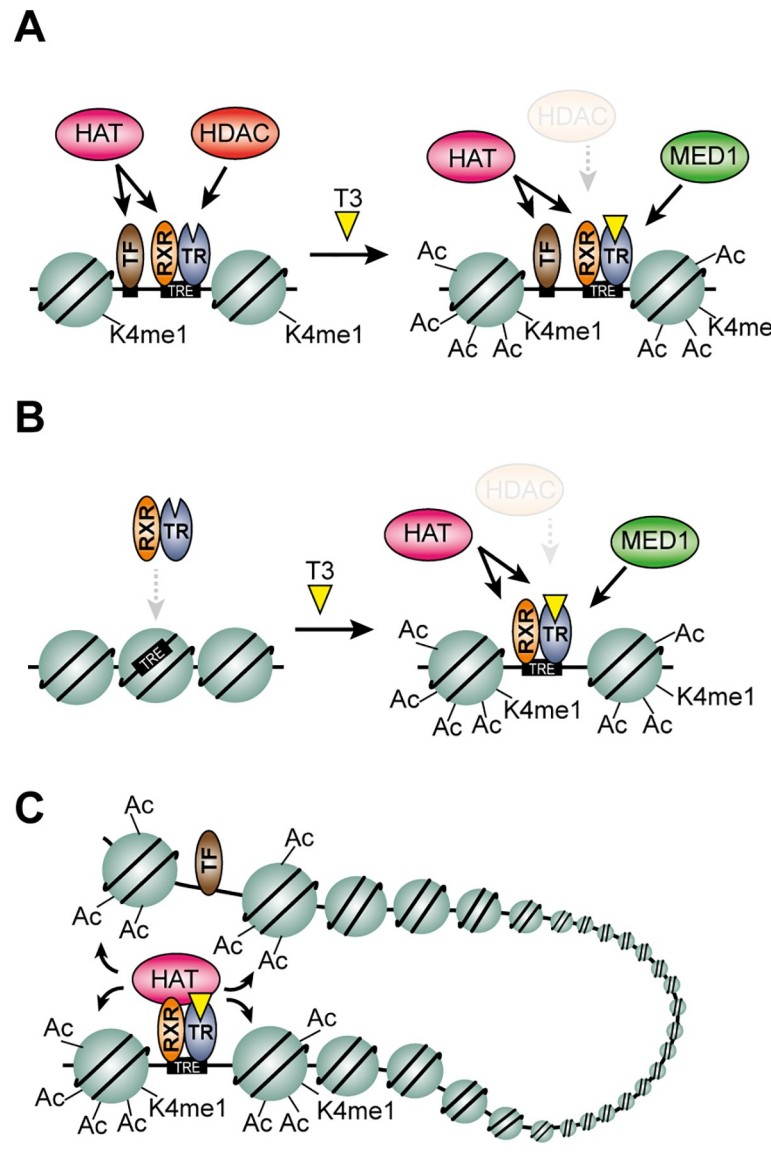

**Fig 6. Models describing T3-dependent H3 hyperacetylation and activation of enhancers.** (A) Poised enhancers (type 1A) are hyperacetylated as a result of HDAC3 disassociation from the TRBS. As HDAC3 occupancy is reduced, histone acetylation increases as a result of pre-occupied HATs. In addition, MED1 is recruited to the enhancers in a T3-dependent manner. (B) Dormant enhancers (type 1B) are hyperacetylated as a result of T3-dependent TR and HAT recruitment to chromatin. Activation of these enhancers is associated with increased chromatin accessibility, H3K4me1 and MED1 recruitment. (C) A subset of T3-activated enhancers is not occupied by TR yet activated in a TR-dependent manner. These enhancers are connected to TR occupied enhancers either though long-range interaction within TADs or local interaction within SEs. The TRE illustrates the TR response element, which predominantly consists of DR4 motifs.

T3, suggesting that these dormant T3-responsive enhancers are established in a hormone-dependent manner (Fig 6B). Since the majority of these enhancers were H3K27Ac within 2h of T3 treatment suggests that these enhancers are directly regulated by T3 dependent recruitment of TR to chromatin. This agrees with recent observations showing that TR can be ligand dependently recruited to a subset of TRBSs and induce *de novo* chromatin accessibility [32,33]. Thus, hypoacetylation of these regions in absence of T3 is simply a consequence of a condensed chromatin environment with low occupancy of transcription factors and HATs. In

agreement, we found that disruption of NCOR1 was less likely to result in increased histone acetylation compared to the poised enhancers. Importantly, correlation with gene expression analysis suggests that a number of T3-induced genes are regulated by enhancers controlled by ligand-dependent enhancer establishment. This agrees with earlier studies showing that a subset of T3-induced genes are not repressed by TR in hypothyroid condition [61].

In addition to the TRBS described above, we identified a number of enhancers hyperacetylated by T3 without any apparent occupancy of TR and presence of canonical TR response elements. Importantly, perturbed TR activity abolished T3-dependent hyperacetylation, demonstrating that these enhancers are regulated in a TR-dependent manner. In agreement, disrupted T3-induced hyperacetylation has also been observed in the liver of TRβ KO mice [62]. Interestingly, disrupted hyperacetylation is also observed in the liver of mice expressing TRβ mutant (TR GS) not able to bind DNA, suggesting that type 1 actions are predominantly involved in T3-mediated hyperacetylation [62]. Moreover, disrupting type 4 actions of TR does not impair histone hyperacetylation [62]. Thus, T3-mediated histone hyperacetylation of regions not occupied by TR is likely a consequence of secondary effects of type 1 actions of TR, including regulation of secondary TFs. We used extensive DNA motif analysis to test if secondary TR-regulated TFs may be involved in T3-dependent hyperacetylation, however, this failed to reveal obvious putative candidates. In contrast, analysis of the higher-order chromatin structure suggests that chromatin organization may be partly involved in T3-dependent hyperacetylation of enhancers without apparent occupancy of TR (Fig 6C). First, a fraction of T3-regulated enhancers, not occupied by TR, is located in SEs containing TRBSs. Second, hyperacetylated enhancers containing TRBS are interacting with a fraction of T3-regulated enhancers lacking TRBS. Using CRISPR-mediated genome editing it has previously been shown that deletion of a specific enhancer can lead to reduced H3K27Ac of interacting enhancers [63], suggesting possible functional hierarchy between interacting enhancers. Future functional studies of T3-regulated enhancers using genome editing technology will likely provide functional insights to communication between interacting T3-regulated enhancers.

The three different models shown in Fig 6 seek to explain different mechanisms controlling T3-induced histone acetylation, yet a direct link to T3-regulated genes remains a challenge. To initially address the question, we used a 100kb-to-TSS approximation approach with the obvious disadvantage of being unable to directly connect every hyperacetylated enhancer with a given gene promoter. We found that less than half of the identified T3-activated genes contained a hyperacetylated region within 100kb of the TSS. This suggests a) that a large proportion of T3-induced gene transcription may be regulated independently of H3 hyperacetylated enhancers or b) that a majority of T3-induced genes are controlled by hyperacetylated enhancers positioned more than 100kb from the TSS. To address this, future high-resolution enhancer-promoter capture HiC experiments will be essential. Interestingly, of the 472 T3-induced genes that were associated with nearby hyperacetylated regions, we found that only a small subset was dysregulated in the NCOR1ΔID mutant. This indicates that the canonical bimodal switch model, where NCoR represses gene expression in hypothyroid condition, only explains regulation of a minority of T3-regulated genes. Similar observations have been reported using NCOR1KO and TRKO models [38,61]. Thus, we suggest that T3-induced gene expression is explained by different mechanisms depending on the chromatin context of involved enhancers. This includes the mechanisms illustrated in Fig 6 and future mapping of the higher order chromatin structure in hypo- and hyperthyroid condition will help delineating regulatory mechanisms for individual T3-regulated genes.

In conclusion, we show that T3-regulated histone acetylation is controlled by a number of putative mechanisms, which do not fully agree with the prevailing bimodal switch model. This argues for revision of the model and provides additional insight to NCoR independent

regulation of T3-responsive enhancers and highlight the complexity of hormone dependent gene regulation that may be relevant for other ligand-regulated transcription factors such as RAR, ER and VDR able to occupy chromatin in absence of agonist.

## Materials and methods

### Ethics sstatement

All animal work was performed in accordance with local rules and approved by the National Cancer Institute Animal Care and Use Committee (LMB-037).

### Mouse models

The L-NCOR1ΔID/ID (NCOR1ΔID) and the TRβPV/PV (TR-PV) mouse models were generated and assessed as described earlier [39,50,64]. L-NCOR1ΔID mice were maintained on a mixed C57BL/6;129S strain background. TR-PV mice and wildtype littermates were maintained on a C57BL/6J;NIH Black Swiss background. Only male mice were used in this study. Mice within each genetic background had similar age, total body and liver weights (S1 Table). Mice were housed in single or group cages at ~23˚C, in a 12h light-dark cycle and had free access to water. To induce hypo- and hyperthyroidism we fed mice an *ad libitum* low iodine diet containing 0.15% PTU (TD.95125, Envigo Teklad) for 16 days leading to hypothyroidism as described previously [65]. Mice were subsequently randomized and injected intraperitoneal with 5µg of T3 (T2752, Sigma-Aldrich) or vehicle for the last 4–6 days resulting in a hypothyroid and hyperthyroid population [65]. On the last day of the T3 treatment protocol mice were sacrificed and livers were isolated and snap-frozen in liquid nitrogen. In the acute treatment setup hypothyroid mice were injected with a single dose of 5µg of T3 and sacrificed 2 and 6 hours later. On the last day of the T3 treatment protocol mice were sacrificed, and livers were isolated and snap-frozen in liquid nitrogen. The short-term treatment of PTU and PTU and T3 had no apparent effects on mouse body weight and liver weight (S1 Table).

### Chromatin immunoprecipitation

MED1, HDAC3, NCOR1, CBP, p300, SRC-1, H3K27Ac, H3K9Ac and H3K4me1 ChIP experiments were essentially performed as described in [66]. ~25mg of frozen liver material was used per IP from 3–4 mice in each group (WT PTU, WT T3, TR-PV PTU, TR-PV T3 or NCOR1ΔID/ID PTU). Liver samples were homogenised using IKA ULTRA-TURRAX in crosslinking agent. In MED1, HDAC3, CBP, p300, SRC-1 and NCOR1 ChIP experiments samples were crosslinked in 2.1mM DSG (c1104, Proteochem) for 30min at RT with rotation mediating protein-protein fixation before 10min crosslinking in 1% formaldehyde for protein-chromatin fixation. Crosslinking was quenched in 0.125M glycine, washed 2x in cold 1xPBS and resuspend in 1ml lysis buffer (0.1% SDS, 1% Triton X-100, 0.15M NaCl, 1mM EDTA, 20mM Tris-HCl pH8.0, 1x Complete Protease Inhibitor Cocktail (04693116001, Roche) and 1µg/µl BSA). Chromatin was sonicated using Diagenode Biorupter Twin or Covaris ME220 to shear DNA into fragments of ~200-1000bp in size. An initial 30min pre-clearing step with 30µl Protein A/G beads was performed before IP. 2µg/IP of antibody was used in all ChIP experiments except in H3K27Ac and NCOR1 ChIPs where 0.2µg/IP and 3µl/IP whole antiserum was used, respectively. Antibodies used: HDAC3 (sc-11417), CBP (sc-369), p300 (sc-584), SRC-1 (sc-8995), MED1/TRAP220 (sc-8998) from Santa Cruz Biotechnology, NCoR (ab24552), H3K27Ac (ab4729), H3K4me1 (ab8895) from Abcam and H3K9Ac (06–942) from Millipore. Samples were incubated ON at 4˚C with rotation followed by addition of 30µl Protein A/G PLUS-Agarose beads (sc-2003, Santa Cruz Biotechnology) and an additional 2.5-3h

incubation. The immunoprecipitated samples were washed and DNA was eluted by 30min incubation at RT with rotation in elution buffer (1% SDS and 0.1M NaHCO$_3$). Reverse-cross-linking was mediated by addition of 5M NaCl followed by ON incubation at 65˚C. DNA was purified with phenol-chloroform extraction and precipitated with ethanol. Recovery and quality were evaluated by qPCR (see S2 Table for primers used), before sample pooling, library preparation and sequencing.

## RNA purification

~10-15mg of frozen liver material was used for RNA isolation from 4 biological replicates. The samples were homogenized, and RNA extracted using TRI-Reagent (T9424, Sigma-Alrich) and EconoSpin Columns (1920–250, Epoch Life Science) according to manufacturer's protocol. RNA concentration was determined by NanoDrop ND-1000 Spectrophotometer. RT-qPCR was performed to verify expected expression level of selected hypo- and hyperthyroid responsive genes. RNA quality was determined on Agilent Fragment Analyzer using Standard Sensitivity RNA Analysis Kit (DNF-471, Agilent).

## Library construction and sequencing

Libraries were prepared using NEBNext reagents and kits for Illumina Sequencing following manufacturer's instructions. In brief, mRNA was isolated from 1µg total RNA using magnetic poly d(T) beads (E7490L). cDNA was synthesized using RNase inhibitor (M0314L), Reverse Transcriptase (M0368L) and buffers, random primers, enzymes from NEBNext Ultra II kit (E7771L). DNA and cDNA samples were end-repaired and 5' phosphorylated using a mix of T4 ligase buffer (B0202S), dNTP (N0447S), T4 DNA polymerase (M0203L), Klenow DNA polymerase (M0210S) and T4 polynucleotide kinase (M0201S) followed by 30min incubation at RT. Samples were cleaned and size-selected using magnetic AMPure XP beads (A63881, Beckman Coulter). 3'-ends were adenylated by a 30min 37˚C incubation in A-tailing mix containing NEBuffer 2 (B7002S), dATP (N0440S) and Klenow exo (3'->5' exo-) (M0212S). 5'-phosphylated and 3'-adenylated DNA ends allow hairpin-adapter ligation using Blunt/TA ligase (M0367L) incubating 15min at RT. Adaptors were cleaved by USER enzyme (E6609S, E6440S). Samples were cleaned with magnetic AMPure XP beads. Libraries were amplified with 10–16 cycles of PCR using Q5 polymerase (B9027S) and index primers (E6609S, E6440S). Clean up and size selection steps were performed using AMPure XP magnetic beads. Library concentration was determined using KAPA library Quantification Kit (KK4854, Kapa Biosystems) following manufacturer's instructions. Libraries were sequenced on an Illumina HiSeq 1500 or Illumina NovaSeq 6000 instrument.

## Sequencing data analysis

*Sequencing*: Sequenced data was aligned to the reference mouse genome mm9 using STAR [67] and aligned tags were normalized and quantified using HOMER [4]. Number aligned reads and uniquely mapped reads for each library can be found in S3 Table. Sequenced data was visualized using the UCSC Genome Browser.

*ChIP-seq analysis*: T3-responsive histone hyperacetylated regions were identified using HOMER [4] by findPeaks –region–size 1000 and –MinDist 2500. Differential T3-regulated histone acetylated regions were identified with DESeq2 [68] using the thresholds FDR<0.01 and log2FC>1. High-confidence hyperacetylated regions were defined as regions significantly hyperacetylated at both H3K27 and H3K9. Previous published DNase-seq data [32] was used to identify putative transcription factor binding sites within the histone acetylated regions. T3-induced hyperacetylated regions bound by TR were ranked based on HDAC3 occupancy

in the hypothyroid condition. The top 1/3 of the regions, with high HDAC3 occupancy (type 1A TRBS), and the bottom 1/3 of the regions, with low HDAC3 occupancy (type 1B TRBS), were used for further comparison of poised and established enhancers, respectively. Quantification of histone modification, chromatin accessibility and co-regulator recruitment was performed by HOMER and average tag counts from two biological replicates were used for differential tag quantification. HDAC3 ChIP-seq peaks were identified using HOMER by findPeaks -style factor and replicate concordant peaks (from two biological replicates) was used for downstream analysis. SEs were identified based on H3K27Ac levels using HOMER by findPeaks -style super. Heatmaps were generated using the pheatmap package in R or MeV [69]. Random genomic regions were selected using Sample() in R.

*DNA motif analysis*: *De novo* motif analysis was performed using HOMER by findMotifs-Genome using 200bp regions extracted from DHS within hyperacetylated regions. IMAGE [42] identified potential transcription factors regulated by TR involved in T3-regulated histone hyperacetylation. Aligned H3K27Ac or H3K9Ac ChIP-seq and RNA-seq data was used as input for IMAGE. Motif activity scores with $p<0.01$ and differential motif activity (T3 vs PTU) $>0$ were extracted for further analysis. Motif similarity was analysed by hierarchical clustering of pearson correlation of the position weight matrix of individual motifs and plotted as a heatmap using R (Pheatmap). Motif scores of DNA motifs enriched by IMAGE analysis and *de novo* motif analysis were quantified using HOMER by annotatePeaks using -mscore setting.

*HiC analysis*: Illumina paired end sequencing data from GSE104129 [46] was aligned by STAR (mm9 genome assembly). Aligned data from two different timepoints of a circadian rhythm (ZT10 and ZT22) was combined and genomic interactions were identified using HOMER by findTADsAndLoops with a resolution of 5000bp and a window size of 10000bp. Similarly, quantification of interaction between specific regions was determined by HOMER by analyzeHiC using a resolution of 5000bp and a window size of 10000bp. Statistical difference between no/TRBS regions and w/TRBS regions interactions compared to regions in the opposite direction (control) was determined by Wilcoxon Signed Rank Test. T3 co-regulated regions confined in TADs were determined using available TAD information from mouse liver tissue (Kim et al., 2018). Random genomic regions were selected using Sample() in R.

*RNA-seq analysis*: Differential gene expression was identified with DESeq2 using the threshold FDR$<0.01$. Furthermore, T3-induced genes were defined as log2FC$>1$ (T3 vs PTU). The analysis was further focused on genes with a nearby hyperacetylated regions with and without occupancy of HDAC3 ($\pm$100kb). Differentially regulated genes in NCOR1$\Delta$ID mice was defined with FDR$<0.01$ and log2FC$>0$. Random genes were selected using Sample() in R.

*Data visualization*: All Venn diagrams, volcano plots, heatmaps and histograms were generated using R.

*Analysed NGS data availability*: ChIP-seq count tables, ChIP/HiC/SE genomic positions and RNA expression data can be found in S1–S6 Data.

## qPCR data analysis

Microsoft Office Excel software (v16.16.10) was used for analysing statistical parameters for qPCR data, which is presented as mean with SD. Statistical significance was calculated using a two-tailed unpaired Student's t-test. Numerical data can be found in S7 Data.

## Previously published data used for analysis

Previously published TR ChIP-seq and DNase-seq data is from The Sequence Read Archive: SRP055020 (https://www.ncbi.nlm.nih.gov/sra/?term=SRP055020) [32] or Gene Expression

Omnibus: GSE52613 (https://www.ncbi.nlm.nih.gov/geo/query/acc.cgi?acc=GSE52613) [33].
Previously published HiC data is from Gene Expression Omnibus: GSE104129 (https://www.
ncbi.nlm.nih.gov/geo/query/acc.cgi?acc=GSE104129) [46]. Previously published liver
H3K27Ac ChIP-seq data from hypothyroid NCOR1 KO and euthyroid WT mice is from Gene
Expression Omnibus: GSE99475 (https://www.ncbi.nlm.nih.gov/geo/query/acc.cgi?acc=
GSE99475) [38].

## Supporting information

**S1 Fig. Identification of H3K27Ac and H3K9Ac regions in the liver of hypo- and hyperthyroid mice.** (A) Correlation between two replicate H3K27Ac ChIP-seq experiments in mice
treated with PTU. (B) Correlation between two replicate H3K27Ac ChIP-seq experiments in
mice treated with PTU+T3. (C) Correlation between two replicate H3K9Ac ChIP-seq experiments in mice treated with PTU. (D) Correlation between two replicate H3K9Ac ChIP-seq
experiments in mice treated with PTU+T3. (E) Replicate concordant H3K27Ac regions in
mice treated with PTU and PTU+T3. (F) Replicate concordant H3K9Ac regions in mice
treated with PTU and PTU+T3. (G) Combined number of identified H3K27Ac and H3K9Ac
regions. (H) Correlation between H3K27Ac and H3K9Ac in mice treated with PTU. (I) Correlation between H3K27Ac and H3K9Ac in mice treated with PTU+T3. (J) Size of identified
H3K27Ac regions. (K) Size of identified H3K9Ac regions.
(TIF)

**S2 Fig. Identification of H3K27 and H3K9 hyperacetylated regions in the liver of hypo-
and hyperthyroid mice.** (A) Identification of 2882 H3K27 hyperacetylated regions in hyperthyroid condition (FDR<0.01 and log2FC>1). (B) Identification of 1928 H3K9 hyperacetylated regions in the hyperthyroid condition (FDR<0.01 and log2FC>1). (C) Correlation
between H3K27- and H3K9 hyperacetylation at 1592 H3 hyperacetylated regions. (D) Correlation between H3K4me1 and H3K27Ac at hyperacetylated regions (n = 1592). (E) Correlation
between H3K4me1 and H3K9Ac at hyperacetylated regions (n = 1592). (F) Distribution of
hyperacetylated regions within exons, introns, promoters and intergenic regions. (G) Correlation between H3K27Ac in hyperthyroid and euthyroid condition (n = 1592). Correlation coefficient (Pearson) indicated in plots panels C, D, E and G. (H) Quantification of H3K27Ac and
H3K9Ac at regions hyperacetylated with (w/TRBS) and without TRBS (no/TRBS) in response
to T3. Statistical difference was determined by a Wilcoxon Signed Rank Test, ***p<0.001.
(TIF)

**S3 Fig. H3K27Ac in response to 2 hour (h) and 6h T3 treatment.** (A) H3K27Ac after 2h of
T3 treatment was quantified at hyperacetylated regions with a TRBS (w/TRBS) and analysed
by DESeq2. FDR<0.05 are coloured red. (B) Quantification of H3K27Ac in response to 2h
and 6h treatment with T3 at hyperacetylated regions with a TRBS. ChIP-seq tag counts are
normalized by a z-score. (C) Percentage of hyperacetylated regions with significant increased
H3K27Ac (FDR<0.05, Log2FC>0) after 2h and 6h treatment with T3. (D) H3K27Ac after 2h
of T3 treatment was quantified at hyperacetylated regions without a TRBS (no/TRBS) and analysed by DESeq2. FDR<0.05 are coloured red. (E) Quantification of H3K27Ac in response to
2h and 6h treatment with T3 at hyperacetylated regions without a TRBS. ChIP-seq tag counts
are normalized by a z-score.
(TIF)

**S4 Fig. Analysis of DNA motifs contributing to hyperacetylation of H3K27 and H3K9.** (A)
Motifs contributing to T3-regulated H3K27Ac. Motifs contributing to T3-induced H3K27Ac
with p<0.01 are coloured yellow. (B) Motifs contributing to T3-regulated H3K9Ac. Motifs

contributing to T3-induced H3K9Ac with p<0.01 are coloured green. (C) Motifs contributing to both H3K27 and H3K9 hyperacetylation by T3. (D) Hierarchical clustering of pearson correlation of the positions weight matrix (PWM) of motifs contributing to T3-induced H3K27 and H3K9 acetylation. Motifs resembling DR4 or DR4 half sites are shown on the right. (E, left) Motifs contributing to T3-induced H3K9Ac and H3K27Ac evaluated by IMAGE analysis. Motifs enriched at p<0.01 are ranked according to the mean differential motif activity (z-score) in response to T3 (Dmotif activity). Normalized motif activities for H3K9Ac and H3K27Ac in hypo- and hyperthyroid condition are visualized as a heatmap. Motifs resembling DR4 or DR4 half site are marked red. (E, right) Statistical test of differential motif scores of hyperacetylated enhancers with and without TRBS. The test was performed using Wilcoxon Signed Rank Test corrected for multiple testing using Benjamini & Hochberg method. (F) De novo DNA motif analysis of DHSs associated with hyperacetylated regions with and without TRBS. Left part of the panel shows statistical test of differential motif scores. The statistical test was performed using Wilcoxon Signed Rank Test corrected for multiple testing using Benjamini & Hochberg method.
(TIF)

**S5 Fig. Interaction between T3-regulated enhancers in mouse liver tissue.** (A) Distance between all interacting regions identified from HiC. (B) Distance between hyperacetylated regions associated with and without TRBSs. (C and D) Examples of interacting regions near T3-regulated genes. Hyperacetylated regions (T3-regulated enhancers) are indicated by green (w/TRBS) and orange (no/TRBS). The T3-regulated *Ginm1* and *Glul* genes are indicated in red.
(TIF)

**S6 Fig. HDAC3 occupancy at TRBS in livers from mice expressing NCORΔID.** (A) Fraction of type 1A and type 1B TRBS with HDAC3 or TR peaks in hypothyroid condition. (B) H3K27Ac at type 1A and type 1B TRBS in response to 2h and 6h T3 treatment. (C) Heatmap illustrating HDAC3 occupancy at TRBS in the NCORΔID mutant compared to WT. TRBS are ranked according to HDAC3 occupancy in hypothyroid WT mice. HDAC3 ChIP-seq performed on livers from hypothyroid animals. (D) Quantification of HDAC3 occupancy at TRBSs associated with type 1A and type 1B TRBSs. Statistical difference was determined by a Wilcoxon Signed Rank Test, ***p<0.001.
(TIF)

**S1 Table. Age, body weight and liver weight of animals.** Data represent average with indicated standard deviations.
(PDF)

**S2 Table. ChIP-qPCR primers.** Primers used in HDAC3, NCOR1, CBP, SRC1, MED1 and p300 ChIP-qPCR experiments presented in Fig 3E.
(PDF)

**S3 Table. Summary of next generation sequencing data.** Sequenced tags were aligned to mm9 using STAR. Uniquely aligned reads were used for downstream analysis.
(PDF)

**S1 Data. NGS count tables for H3 acetylated domains.** Averaged sequence tags from biological replicates were normalized to sequencing depth. Output from DEseq2 analysis is included as basemean, log2FC and FDR. Overlap with TRBS, HDAC3 and hyperacetylated regions is indicated with 0/1.
(XLSX)

**S2 Data. NGS count tables for TRBS.** Averaged sequence tags from biological replicates were normalized to sequencing depth. Overlap with TRBS and HDAC3 is indicated with 0/1 or 1A, 1B (for TRBS).
(XLSX)

**S3 Data. NGS count tables for HDAC3.** Averaged sequence tags from biological replicates were normalized to sequencing depth. Overlap with TRBS is indicated with 0/1.
(XLSX)

**S4 Data. Genomic position of SEs.** SE slope is indicated.
(XLSX)

**S5 Data. HiC contacts.** Overlap with histone hyperacetylated regions and TRBS is indicated with 0/1.
(XLSX)

**S6 Data. Gene expression data (RNA-seq).** Averaged sequence tags from biological replicates were normalized to sequencing depth. Output from DEseq2 analysis is included as log2FC and FDR. Overlap with nearby occupancy of TRBS and histone hyperacetylation is indicated with 0/1.
(XLSX)

**S7 Data. ChIP-qPCR.** Recovery for individual replicates at nine different genomic positions.
(XLSX)

## Author Contributions

**Conceptualization:** Stine M. Præstholm, Lars Grøntved.

**Data curation:** Lars Grøntved.

**Formal analysis:** Stine M. Præstholm, Lars Grøntved.

**Funding acquisition:** Lars Grøntved.

**Investigation:** Stine M. Præstholm, Majken S. Siersbæk, Ronni Nielsen, Xuguang Zhu.

**Methodology:** Anthony N. Hollenberg, Sheue-yann Cheng, Lars Grøntved.

**Project administration:** Lars Grøntved.

**Resources:** Anthony N. Hollenberg, Sheue-yann Cheng.

**Supervision:** Anthony N. Hollenberg, Sheue-yann Cheng, Lars Grøntved.

**Writing – original draft:** Stine M. Præstholm.

**Writing – review & editing:** Majken S. Siersbæk, Anthony N. Hollenberg, Sheue-yann Cheng, Lars Grøntved.

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
