## [Decision Letter · Decision Letter 0]

16 Jan 2020

Dear Dr Grøntved,

Thank you very much for submitting your Research Article entitled 'Multiple mechanisms regulate H3 acetylation of enhancers in response to thyroid hormone' to PLOS Genetics. Your manuscript was fully evaluated at the editorial level and by an independent peer reviewer. Normally, PLOS Genetics prefers to use 3 external reviewers, but despite many invitation we were only able to secure one (most likely due to the busy holiday season rather than interest in the study).  The review we have in hand is thoughtful and thorough so rather than further extend the timeline but seeking additional reviews we've decided to proceed with only one. The reviewer appreciated the attention to an important topic but identified some aspects of the manuscript that should be improved.

The reviewer has several suggestions.  We ask that you pay close attention to making the writing accessible to a wide audience.  In addition we ask that you attend to all the suggestions that can be addressed by textual amendment or additional data analysis.  In some cases the reviewer has suggested additional experimentation – the editors (and reviewer) feel that these could significantly strengthen the manuscript, but ultimately acceptance of the manuscript will not be contingent on adding them (though we strongly encourage you to carefully consider them).

We therefore ask you to modify the manuscript according to the review recommendations before we can consider your manuscript for acceptance. Your revisions should address the specific points made by each reviewer.

[LINK]

Yours sincerely,

Gregory P. Copenhaver

Editor-in-Chief

PLOS Genetics

Wendy Bickmore

Section Editor: Epigenetics

PLOS Genetics

Reviewer's Responses to Questions

**Comments to the Authors:**

Reviewer #1: The article of Praestholm et al is a genome wide analysis of mouse liver chromatin, which aims is to understand the mechanism of action of thyroid hormone (T3) and its nuclear receptors (TRs).

Overall this is a very interesting and very novel study, bringing a lot of information on a basic mechanism of gene regulation, which interest goes beyond the narrow field of T3 endocrinology. The data are quite convincing, well analyzed, and the authors reach interesting interpretations. Although the recurrent use of arbitrary threshold is risky (100 kb between TRBS and transcription start site for example), the conclusions seem reliable.

The main weakness of the study is the text presentation, which I found very difficult to follow. This is due in part to the complexity of the experiments, but the writing is also sometimes obscure. For example, the sentence line 295-299 is very long and I had to read it several times to understand it. Part of the problem is that there no abbreviation for entities like “chromatin regions in which histone acetylation is induced after T3 treatment”, and the authors might consider creating some.

I have a number of questions, and suggestions for additional experiments:

1) The hypothyroid animals were treated for 5 days with T3. This is a very long time, which leaves time for a cascade of indirect effects to take place. I would recommend using a much shorter treatment, perhaps 3 hours only, and ideally to perform a time course experiment. This would help to separate causes from consequences. In particular:

- Fig3D: does TR come first, decompacting chromatin? Is histone methylation first giving access to chromatin? Or is a T3 responsive gene encoding some unknown decompacting factor?

- is the acetylation of regions which are not occupied by TR delayed, compared to the one with a TRBS? This would address the possibility that T3 induces the expression of genes encoding transcription factors able to induce chromatin acetylation at distant sites. This seems to be a good way to challenge the model showed Fig 6C.

2) The authors mention the classical SRC1 histone acetyl transferase coactivator, but did not perform a ChipSeq analysis for this obvious candidate. Is this for technical reason? This gap should at least be acknowledge as a limitation of the study.

3) The authors use two sets of ChipSeq for TRs, coming from different publications. The two datasets contain very different number of TRBS (around 1500 for Grontvedt et al and 20 000-50 000 for Ramaddoss et al , the later using a viral vector to overexpress a tagged receptor). This must affect the analysis, and the authors should explain to what extend it does.

4) ChipSeq evaluate chromatin association, not DNA binding, therefore a TRBS can reflect tethering of TR by multiprotein complexes. Is this visible in super-enhancers or TADs as the models suggest? One would expect that TRBS appear simultaneously (only one with a consensus TRE) after T3 treatment at several locations. I also believe that if tethering happens, one would expect a better contrast on figure 2C, if only TRBS which contain a DR4 element had been considered.

5) Figure S3 motif search: I suppose that if the DR4 is actually enriched, one should also find enrichment for the half-site (6mer), and thus, accidentally, for other half-sites organization. Therefore it seems that there is no reason to believe that other nuclear receptors are actually present. This should be clarified.

6) I suspect that there is a continuum, ( ie no clear threshold) between poised and dormant enhancers (especially because the number of detected TRBS is highly variable from one experiment to the other and dependent on several statistical thresholds). Perhaps the use of other available TR Chipseq datasets (heart, cell line) could help to ask whether TR requires cell-specific factors to access the DR4 elements, and whether this explains why different cell types activate different genes in response to T3.

7) After reading, one has no response to simple questions: how much of the transcriptional response to T3 is explained by the canonical model, and the alternative models presented by the authors? And how much remains unexplained? Previous studies reported different modes of T3 response for different gene clusters (Yen PMID:12776178; PMID:29229863 Hönes PMID:29229863). Can this be related to the present models? Fig5A indicates that only half of the T3 induced genes are associated with hyperacetylation, while Fig5b 30% of these have no proximal TRBS. This leaves only 35% of the T3 induction explained by the presence of a proximal TRBS. In general, the relationship between chromatin status and gene expression should be better discussed.

8) I suspect that if a 100 kb threshold distance is used, as the authors did, most if not all genes might have a proximal TRBS. In that case it is difficult to understand why only 1015 genes are upregulated by T3. Should the TRBS and the gene be in the same TAD? It would be interesting to consider numbers in the other direction: among all the DR4 consensus found in the genome, which fraction is occupied by TR before and after T3 treatment? Among those, which ones are occupied by NcoR in hypothyroid mice, and Med1 is T3 treated mice? Is the T3 response of a gene always accompanied by hyperacetylation of the nearby chromatin? how many genes are close to a gene which is T3 responsive?

I believe that it would be in the authors interest to perform at least some of the suggested experiments. In conclusion, although I found the reading difficult, I am convinced that the authors made a substantial progress in the understanding of gene regulation by T3, in a relevant physiological system, which is a very significant achievement.

Other points:

Line 179 or 767: How is exactly made the random selection?

Line 194: is it possible to make this statement statistically significant?

Fig 2F: subTAD is not defined and the interpretation obscure. This might require further explanation.

FigS3F1: I believe that this is DR4 not DR1

Fig 2H: I do not really followed the explanation in the legend and the text. Perhaps this could be clarified.

Fig 3C: Unless I missed it, the legend or the method section do not explain how this heatmap was generated.

Fig 3E: the genomic coordinates of the amplified fragments should be given somewhere.

Fig 4H: the genomic coordinates of these regions should be indicated in the legend (and the possible presence of genes).

**Have all data underlying the figures and results presented in the manuscript been provided?**

Reviewer #1: Yes

PLOS authors have the option to publish the peer review history of their article (what does this mean?). If published, this will include your full peer review and any attached files.

Reviewer #1: No

---

## [Editor Report · Decision Letter 1]

8 Apr 2020

Dear Dr Grøntved,

We are pleased to inform you that your manuscript entitled "Multiple mechanisms regulate H3 acetylation of enhancers in response to thyroid hormone" has been editorially accepted for publication in PLOS Genetics. Congratulations!

Yours sincerely,

Gregory P. Copenhaver

Editor-in-Chief

PLOS Genetics

Comments from the reviewers (if applicable):

**Data Deposition**

http://datadryad.org/submit?journalID=pgenetics&manu=PGENETICS-D-19-01773R1

**Press Queries**

---

## [Editor Report · Acceptance letter]

15 May 2020

PGENETICS-D-19-01773R1 

Multiple mechanisms regulate H3 acetylation of enhancers in response to thyroid hormone 

Dear Dr Grøntved, 

We are pleased to inform you that your manuscript entitled "Multiple mechanisms regulate H3 acetylation of enhancers in response to thyroid hormone" has been formally accepted for publication in PLOS Genetics! Your manuscript is now with our production department and you will be notified of the publication date in due course.

With kind regards,

Laura Mallard

PLOS Genetics

On behalf of:
